# BRAINLM:
# A FOUNDATION MODEL FOR BRAIN ACTIVITY RECORDINGS

Josue Ortega Caro[*,1], Antonio H. de O. Fonseca[*,2], Syed A. Rizvi[*,4], Matteo Rosati[*,5], Christopher Averill[3], James L. Cross[4], Prateek Mittal[4], Emanuele Zappala[11], Rahul M. Dhodapkar[7], Chadi G. Abdallah[3, **], David van Dijk[1,2,4,8,9,10, **],

[1]Wu Tsai Institute, [2]Interdepartmental Neuroscience Program, Yale University, [3]Baylor College of Medicine, [4] Department of Computer Science, [5] Yale School of Medicine, [7] University of Southern California, [8] Interdepartmental Program in Computational Biology & Bioinformatics, [9] Internal Medicine, [10] Cardiovascular Research Center, Yale University,[11] Department of Mathematics and Statistics, Idaho State University, [*] Co-first authors, [**] Co-Corresponding authors

## ABSTRACT

We introduce the Brain Language Model (BrainLM), a foundation model for brain activity dynamics trained on 6,700 hours of fMRI recordings. Utilizing self-supervised masked-prediction training, BrainLM demonstrates proficiency in both fine-tuning and zero-shot inference tasks. Fine-tuning allows for the accurate prediction of clinical variables like age, anxiety, and PTSD as well as forecasting of future brain states. Critically, the model generalizes well to entirely new external cohorts not seen during training. In zero-shot inference mode, BrainLM can identify intrinsic functional networks directly from raw fMRI data without any network-based supervision during training. The model also generates interpretable latent representations that reveal relationships between brain activity patterns and cognitive states. Overall, BrainLM offers a versatile and interpretable framework for elucidating the complex spatiotemporal dynamics of human brain activity. It serves as a powerful "lens" through which massive repositories of fMRI data can be analyzed in new ways, enabling more effective interpretation and utilization at scale. The work demonstrates the potential of foundation models to advance computational neuroscience research.

## 1 INTRODUCTION

Understanding how cognition and behavior arise from brain activity stands as one of the fundamental challenges in neuroscience research today. Functional magnetic resonance imaging (fMRI) has emerged as a critical tool for pursuing this goal by providing a noninvasive window into the working brain. fMRI measures blood oxygen level fluctuations that reflect regional neural activity. However, analyzing the massive, high-dimensional recordings produced by fMRI poses major challenges. The blood-oxygen-level dependent (BOLD) signals represent an indirect measure of brain function and can be difficult to interpret. Furthermore, fMRI data exhibits complex spatiotemporal dynamics, with critical dependencies across both space and time. Most existing analysis approaches fail to fully model these complex nonlinear interactions within and across recordings (Seewoo et al., 2021).

Prior fMRI analysis techniques have relied heavily on machine learning models designed for specific narrow tasks (Takagi & Nishimoto, 2023; Mozafari et al., 2020; Ozcelik et al., 2022), hindering their generalizability. Traditional models also struggle to integrate information across the wealth of unlabeled fMRI data available. Hence, there is an ongoing need for flexible modeling frameworks that can truly capitalize on the scale and complexity of fMRI repositories.

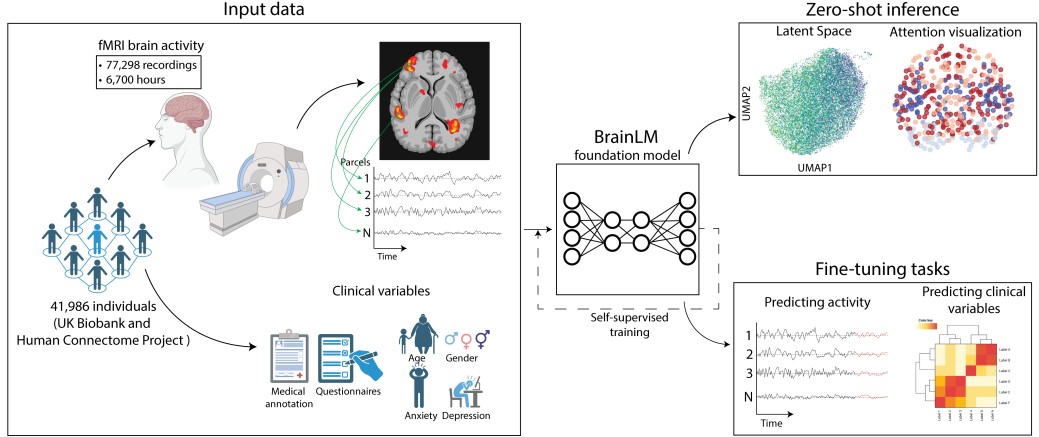

Figure 1: Overview of the BrainLM framework. The model was pretrained on 77,298 fMRI samples, summing to 6,700 hours of fMRI recordings. BrainLM is pretrained via spatiotemporal masking and reconstruction. After pretraining, BrainLM supports diverse capabilities through fine-tuning and zero-shot inference. Fine-tuning tasks demonstrate the prediction of future brain states and clinical variables from recordings. Zero-shot applications include inferring functional brain networks from attention weights. This highlights BrainLM's versatility as a foundation model for fMRI analysis.

Foundation models represent a new paradigm in artificial intelligence, shifting from narrow, task-specific training to more general and adaptable models (Brown et al., 2020). Inspired by breakthroughs in natural language processing, the foundation model approach trains versatile models on broad data at scale, enabling a wide range of downstream capabilities via transfer learning. Unlike previous AI systems designed for singular functions, foundation models exhibit general computational abilities that make them suitable for myriad real-world applications. Large language models like GPT have demonstrated the potential of this framework across diverse domains including healthcare, education, robotics, and more (Bommasani et al., 2021; Wiggins & Tejani, 2022; Orr et al., 2022; Mai et al., 2023). Foundation models offer new opportunities to rethink challenges in neuroscience and medical imaging analysis.

Here, we introduce BrainLM, the first foundation model for fMRI recordings. BrainLM leverages a Transformer-based architecture to capture the spatiotemporal dynamics inherent in large-scale brain activity data. Pretraining on a massive corpus of raw fMRI recordings enables unsupervised representation learning without task-specific constraints (see Figure 1). After pretraining, BrainLM supports diverse downstream applications via fine-tuning and zero-shot inference. We demonstrate BrainLM's capabilities on key tasks including prediction of future brain states, decoding cognitive variables, and discovery of functional networks. Together, these results highlight BrainLM's proficiency in both zero-shot and fine-tuning tasks. This work highlights the potential of applying large language models to advance neuroscience research. BrainLM is a foundation model for the community to build upon, providing more powerful computational tools to elucidate the intricate workings of the human brain.

## 2  RELATED WORK

Prior work has explored various machine-learning techniques for analyzing fMRI recordings. Earlier approaches focused on decoding cognitive states from activity patterns. Methods like SVM and neural networks were trained in a supervised fashion to classify fMRI data into stimulus categories or regress against variables of interest (Horikawa & Kamitani, 2017; Hoefle et al., 2018; Beliy et al., 2019). However, these models learn representations tailored to specific tasks and struggle to generalize.

Recent work has aimed to obtain more transferable fMRI encodings without task-specific constraints. Techniques include training autoencoders to reconstruct recordings, learning to map recordings to a lower-dimensional space (Takagi & Nishimoto, 2023; Mozafari et al., 2020; Ozce-

lik et al., 2022). However, most methods operate on small datasets, limiting their ability to learn robust generalizable representations. Our work is most closely related to recent efforts to apply masked autoencoders for unsupervised pretraining on fMRI data (Chen et al., 2023a;b) or other neural recording modalities (Zappala et al., 2022; Fonseca et al., 2023, Ye & Pandarinath, 2021; Kostas et al., 2021). However, these prior papers have focused on using the autoencoder pretraining only for visual cortex, and were trained on 1-2 orders of magnitude fewer data compared to BrainLM. Furthermore, all other fMRI models have only focused on applying the MAE framework for specific applications such as stimuli reconstruction. In contrast, BrainLM is a foundational model of all brain regions and has been trained on substantially more data which allows it to learn more powerful encodings of spatiotemporal fMRI patterns.

In addition, other work has focused on finding representational similarities between large language models and brain recordings (Caucheteux et al., 2022; Pasquiou et al., 2022). Work along this line has found high correlation between LLMs and specific brain areas such as language processing regions. However, this work does not focus on learning foundation models of brain dynamics nor finetuning their models for downstream biological tasks. These two are key differences between their approach and ours.

## 3 METHODS

### 3.1 DATASETS AND PREPROCESSING

We leveraged two large-scale publicly available datasets - the UK Biobank (UKB) (Miller et al., 2016) and the Human Connectome Project (HCP) (Elam et al., 2021). The UKB comprises a robust collection of 76,296 task-based and resting-state functional MRI (fMRI) recordings, accompanied by medical records, from a demographic spanning ages 40 to 69. Recordings were acquired on a Siemens 3T scanner at 0.735s temporal resolution. The HCP contains 1,002 high-quality fMRI recordings from healthy adults scanned at 0.72s resolution.

Our model was trained on 80% of the UKB dataset (61,038 recordings) and evaluated on the held-out 20% and the full HCP dataset. All recordings underwent standard preprocessing including motion correction, normalization, temporal filtering, and ICA denoising to prepare the data (Salimi-Khorshidi et al., 2014, Abdallah, 2021). To extract parcel-wise time series, we parcellated the brain into 424 regions using the AAL-424 atlas (Nemati et al., 2020). This yielded 424-dimensional scan sequences sampled at ~1 Hz. Robust scaling was applied by subtracting the median and dividing by the interquartile range computed across subjects for each parcel. In total, our training dataset comprised 6,700 hours of preprocessed fMRI activity patterns from 77,298 recordings across the two repositories. This large-scale corpus enabled unsupervised pretraining of BrainLM to learn robust functional representations.

### 3.2 MODEL ARCHITECTURE & TRAINING PROCEDURE

Our model is based on a Transformer masked autoencoder structure, drawing inspiration from natural language processing designs like BERT (Devlin et al., 2018) and Vision Transformer (Dosovitskiy et al., 2020; He et al., 2022). At its core, the model is a series of multi-headed self-attention layers that process visible (unmasked) as well as masked patches. The goal is to predict the original signal of the masked patches (refer to Figure 2). Detailed implementation specifics are elaborated upon in the supplementary materials.

During training, we selected random subsequences spanning 200 timesteps from each fMRI recording. These parcel time series were then dissected into blocks of 20 timesteps, leading to 10 non-overlapping segments per subsequence. These segments were transformed into 512-dimensional vectors, and a masking operation was performed at rates of 20%, 75%, or 90%. To facilitate model scaling to hundreds of millions of parameters, we design an alternative masking scheme directly inspired by patch-based image encoding in Computer Vision. We select a random subsequence of 200 timesteps from each fMRI recording, and treat the selected recording window of 424 parcels and 200 timesteps as a 2-dimensional image, where the 424 parcels are ordered by their Y-coordinate in 3D space. This scheme preserves locality of brain regions and allows for multi-parcel encoding of the fMRI recording which reduces the total amount of tokens. Note that parcels tend to be correlated in space, therefore, predicting masked parcels

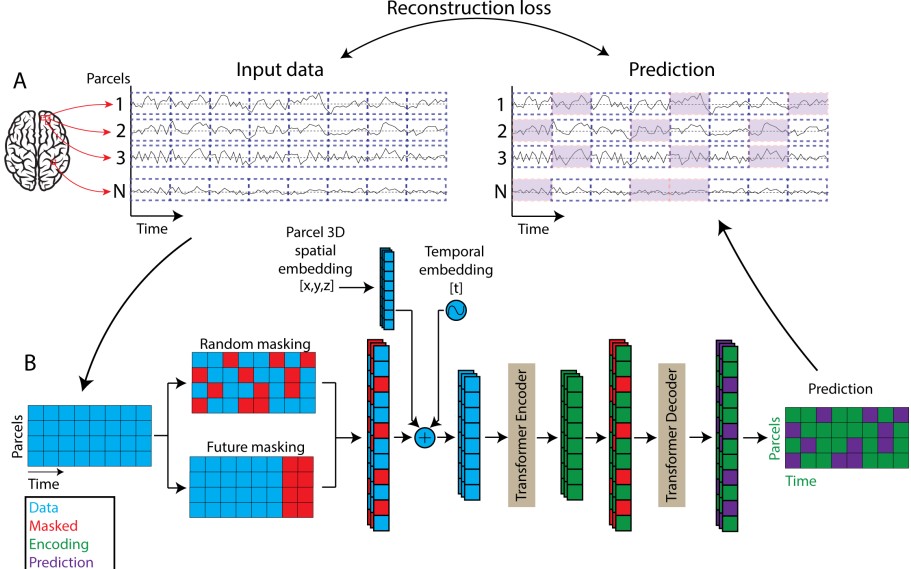

Figure 2: BrainLM architecture and training procedure. A) The fMRI recordings are compressed into 424 dimensions (parcels) (See Methods 3.1). The recordings are randomly trimmed to 200 time points. For each parcel, the temporal signal is split into patches of 20 time points each (blue dashed boxes). The resulting 4240 patches are converted into tokens via a learnable linear projection. B) From the total number of tokens (blue), a subset is masked (red), either randomly or at future timepoints. We then add the learnable spatial and temporal embeddings to each token. These visible tokens (blue) are then processed by a series of Transformer blocks (Encoder). The input to the Decoder is the full set of tokens, consisting of encoded visible tokens (green) and masked tokens (red). The Decoder also consists of Transformer blocks and ultimately projects the tokens back to data space. Finally, we compute the reconstruction loss between the prediction (purple) and the original input data (blue).

within a physical region makes the task significantly more difficult compared to the single-parcel encoding. Only the segments that remained unmasked were processed by a Transformer encoder, which consists of 4 self-attention layers and 4 heads. This was followed by a 2-layer Transformer decoder that processed both the masked and unmasked vectors. The training regimen involved batches of 512 samples, optimized via the Adam algorithm across a span of 100 epochs. The optimization goal was to minimize the mean squared error between the original and the reconstructed signals (visualized in Figure 2). This pretrained model was subsequently leveraged for several tasks including zero-shot brain network inference, fine-tuning for clinical variable prediction, and prognostication of future time states.

## 3.3 CLINICAL VARIABLE PREDICTION

To adapt BrainLM for predicting clinical variables from fMRI recordings, we augmented the pretrained encoder with a 3-layer MLP head. This was then trained to regress targets such as age, neuroticism, PTSD, and anxiety disorder scores. We used Z-score normalization for age. Neuroticism scores were adjusted through min-max scaling, aligning the distribution within the [0, 1] range. For both Post Traumatic Stress Disorder (PCL-5) and General Anxiety Disorder (GAD-7) scores, we first applied a log transformation to moderate their exponential distribution. This was followed by min-max scaling.

We conducted regression on clinical variables using data that BrainLM had not encountered during fine-tuning. For this, we reserved a subset of samples from the UKB test set for both fine-tuning and training SVM regressors. Our evaluation compared the performance of BrainLM with baseline models, using both raw input data and pretrained embeddings. To mitigate overfitting during the fine-tuning process, we introduced a 10% dropout to the activations of both the BrainLM encoder and its MLP head. We also evaluated the pretrained models on zeroshot regression using this same sample subset.

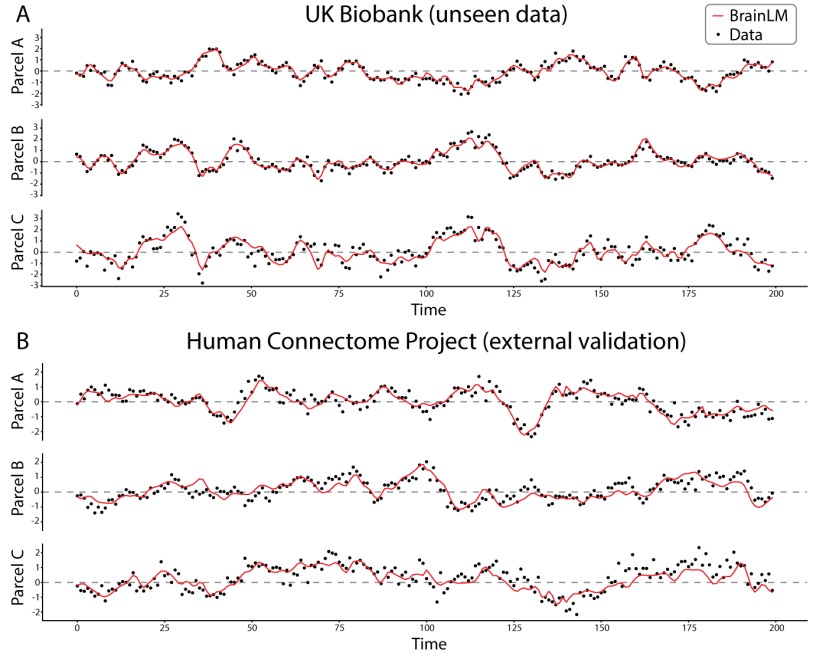

Figure 3: BrainLM reconstruction performance on held-out data. The model predictions (red) closely fit the ground truth recordings (black) of unseen data sampled from the cohort that the model was trained on (A, UKB) as well as data sampled from an external never-before-seen cohort (B, HCP). This demonstrates BrainLM's ability to generalize across subjects and datasets.

## 4 RESULTS

### 4.1 MODEL GENERALIZATION

To evaluate BrainLM's ability to generalize to new fMRI data, we tested its performance on held-out recordings from both the UKB test set as well as the independent HCP dataset.

On the UKB test data, BrainLM achieved an average $R^2$ score of 0.464 on the prediction of masked patches, indicating strong generalization on unseen recordings from the same distribution. Critically, BrainLM also generalized well to the HCP dataset, achieving a $R^2$ score of 0.278. Despite differences in cohort and acquisition details, BrainLM could effectively model brain dynamics in this entirely new distribution. Figure 3 shows sample reconstructions on both UKB and HCP recordings. We found that the model generalization scales with data size and model size, showing strong model scaling (see Figure 4). Overall, these results demonstrate BrainLM's ability to learn robust representations of fMRI recordings that generalize across datasets. This highlights the advantages of large-scale pretraining for learning widely applicable models of brain activity. We will make all pretrained models available via Hugging Face.

### 4.2 PREDICTION OF CLINICAL VARIABLES

One of the primary advantages of foundation models lies in their capability to fine-tune for specific downstream tasks, capitalizing on pretrained representations. An examination of the latent space reveals that our pretrained BrainLM model adeptly encodes information crucial to the clinical variables tied to fMRI recordings, as showcased in Table 14 and Figure 5. To delve deeper into BrainLM's clinical predictive capacity, we adapted the model to regress metadata variables from the UKB dataset. This involved enhancing the pretrained encoder with an MLP head and fine-tuning it to forecast variables such as age, neuroticism, PTSD, and anxiety disorder scores. For this fine-tuning stage, we utilized a subset of the UKB samples that had been set aside and were untouched during the training process. We then compared BrainLM's performance against other prevalent methodologies used in predicting clinical variables, referencing studies like Drysdale et al. (2017); Iidaka (2015). For a benchmark, we factored in the results from an SVM trained on raw data to predict these clinical outcomes. Impressively, across all the measured variables,

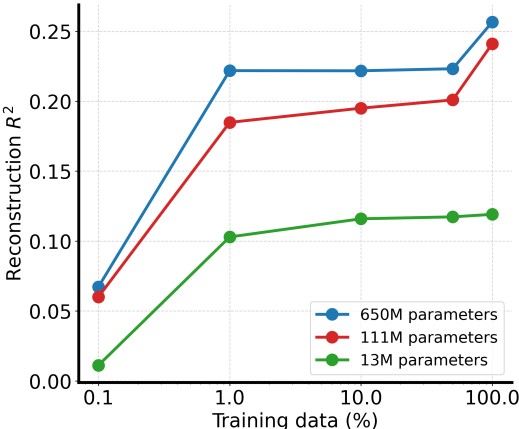

Figure 4: Reconstruction performance on masked fMRI recording patches at varying amounts of training data and model parameter sizes. Increasing model size and data scale consistently yields better performance at self-supervised reconstruction of signal information.

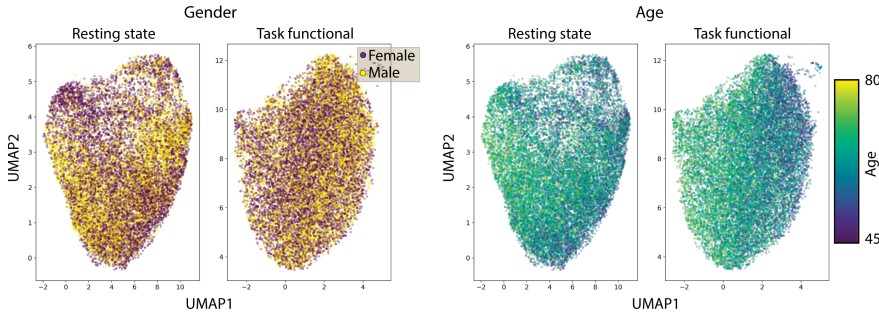

Figure 5: BrainLM learns a latent space encoding clinically relevant information from fMRI recordings. For each fMRI, the CLS token is extracted as a summary representation. The collection of CLS tokens is then embedded into a 2D space using UMAP. The resulting embedding demonstrates organization based on gender (left) and age (right) of the subjects.

BrainLM consistently outperformed other methods, registering a notably lower mean squared error than the competing approaches, as detailed in Table 1. Additional gains from fine-tuning further demonstrate the benefits of initializing with a meaningfully pretrained model before tuning to a specific prediction task. We further explored the quality of the models' recording embeddings by assessing their zeroshot metadata regression performance. As seen in Supp. Table 17, performance scales with model size, indicating that larger models learn more clinically significant information during pretraining.

Overall, these results validate BrainLM's ability to uncover predictive signals within complex fMRI recordings. By leveraging large-scale pretraining and transfer learning, BrainLM offers a powerful framework for fMRI-based assessment of cognitive health and neural disorders. Ongoing work is exploring clinical prediction across a broader range of psychiatric, neurological, and neurodegenerative conditions.

## 4.3 PREDICTION OF FUTURE BRAIN STATES

To evaluate whether BrainLM can capture spatiotemporal dynamics, we assessed its performance in extrapolating to future brain states. A subset of the UKB data was used to fine-tune the model to predict parcel activities at future time points. During fine-tuning, BrainLM was given 180 time-step sequences and trained to forecast the subsequent 20 time steps. We compared

Table 1: Results for the regression of the clinical variables. LSTM, GCN (Kipf & Welling, 2016) and BrainLM were trained with fMRI recordings as input. SVR and MLP used the correlation matrix of parcels as input into the model (see Drysdale et al. (2017) and Iidaka (2015) for more details). The values show the MSE (mean ± std).

|  | Age | PTSD (PCL-5) | Anxiety (GAD-7) | Neuroticism |
|---|---|---|---|---|
| Raw data | 2.0 ± 0.2219 | 0.034 ± 0.0027 | 0.172 ± 0.0066 | 0.160 ± 0.0137 |
| SVR | 0.659 ± 0.036 | 0.022 ± 0.004 | 0.090 ± 0.010 | 0.087 ± 0.008 |
| MLP | 0.693 ± 0.001 | 0.021 ± 0.0003 | 0.081 ± 0.001 | 0.079 ± 0.0005 |
| LSTM | 0.596 ± 0.040 | 0.019 ± 0.001 | 0.083 ± 0.0022 | 0.076 ± 0.002 |
| GCN | 0.862 ± 0.09 | 0.021 ± 0.002 | 0.083 ± 0.02 | 0.077 ± 0.006 |
| BrainLM 13M | **0.464 ± 0.0252** | 0.018 ± 0.0008 | 0.074 ± 0.0053 | 0.072 ± 0.0049 |
| BrainLM 111M | 0.503 ± 0.0207 | **0.015 ± 0.0003** | **0.073 ± 0.0031** | **0.069 ± 0.0038** |

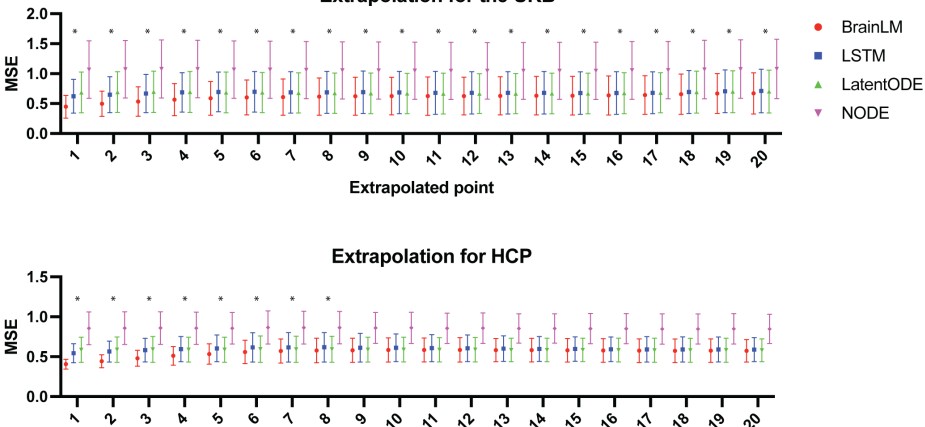

Figure 6: BrainLM outperforms other models in extrapolating future brain states. Models were trained to predict parcel activity 20 timesteps beyond observed context data. The plot shows the mean squared error per timestep on held-out UKB and HCP recordings. BrainLM demonstrates significantly lower error in forecasting near-future brain states. This highlights how pretraining enables BrainLM to effectively learn spatiotemporal fMRI dynamics. The time points for which BrainLM has significantly ($p < 0.05$) lower error than the other methods are identified with "*".

against baseline models including LSTMs (Hochreiter & Schmidhuber, 1997), NODE (Chen et al., 2018; Rubanova et al., 2019), and a non-pretrained version of BrainLM.

As depicted in Figure 6, the fine-tuned BrainLM model significantly surpassed other methodologies when predicting future activity, demonstrating superior performance on both the UKB and HCP test sets. The version of BrainLM without pretraining exhibited a noticeable dip in performance, underscoring the value of pretraining for such tasks. The optimized BrainLM consistently and significantly recorded the least error across all predicted timesteps for the UKB data. For HCP this was significant for the initial 8 timesteps, as detailed in Figure 6. This highlights BrainLM's robust capability to intuitively grasp the dynamics of fMRI.

## 4.4 INTERPRETABILITY VIA ATTENTION ANALYSIS

A key feature of BrainLM is its interpretability. Through the visualization of self-attention weights, we can glean deeper insights into the model's internal representations. We calculated the average attention the BrainLM's CLS token allocated to each parcel during the encoding of fMRI recordings. As highlighted in Figure 7, task recordings demonstrated a pronounced focus on the visual cortex compared to the resting state. This aligns seamlessly with the visual stimuli introduced during tasks. Furthermore, we observed distinctions in attention patterns when

Table 2: Quantitative evaluation of extrapolation performance. Models were tasked with forecasting parcel activity 20 timesteps beyond observed data from the UKB dataset. We compare BrainLM to NODE (Chen et al., 2018), LatentODE (Rubanova et al., 2019) and LSTM. BrainLM shows the best performance across all metrics: higher ($R^2$) and Pearson correlation coefficients ($R$), and lower mean squared error ($MSE$) between predicted and true future states. † MSEs for the larger BrainLM models are not comparable to other models given different normalization of recordings.

|  | UKB | | | HCP | | |
|---|---|---|---|---|---|---|
|  | $R^2$ | $R$ | $MSE$ | $R^2$ | $R$ | $MSE$ |
| BrainLM 650M (fine-tuned) | **0.098** | **0.313** | **0.019**† | **0.061** | **0.253** | **0.018**† |
| BrainLM 111M (fine-tuned) | 0.095 | 0.309 | 0.020† | 0.056 | 0.244 | 0.018† |
| BrainLM 13M (fine-tuned) | 0.086 | 0.280 | 0.648 | 0.028 | 0.185 | 0.568 |
| Transformer 13M (w/o pre-training) | 0.012 | 0.112 | 0.695 | 0.007 | 0.090 | 0.583 |
| LSTM | -0.001 | 0.151 | 0.704 | -0.020 | 0.049 | 0.598 |
| Neural ODE | -0.577 | 0.001 | 1.083 | -0.469 | 2.010e-4 | 0.857 |
| Latent ODE | 0.001 | 0.023 | 0.703 | -0.003 | -2.026e-4 | 0.588 |

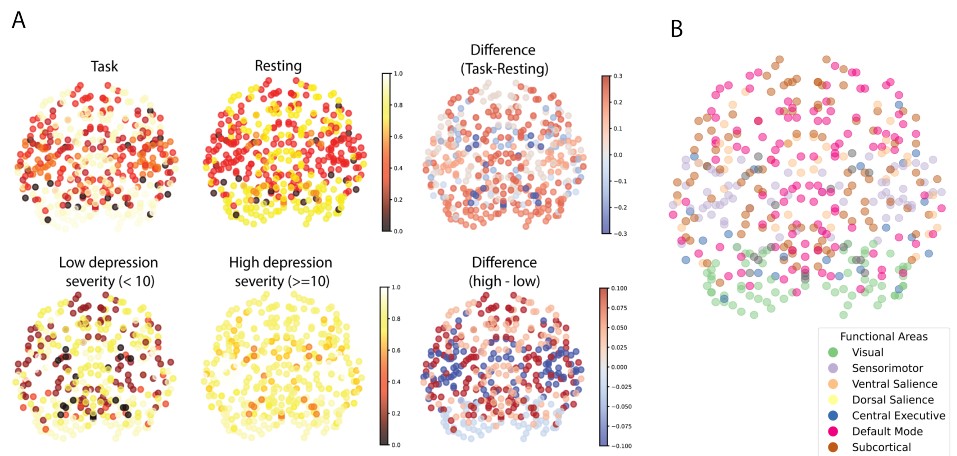

Figure 7: BrainLM attention maps reveal functional contrasts. A) Differences in parcel attention weights between task (left) and rest (right), and low and high depression scores (PHQ9). Task vs. resting state differences highlight changes in the visual cortex. Comparing the severity of depression, the difference highlights subcortical areas. B) Attention can localize parcels to 7 functional networks without supervision. This demonstrates BrainLM's ability to learn meaningful representations in an unsupervised manner.

comparing mild to severe depression, as indicated by the PHQ-9 scores. Notably, there was a pronounced emphasis on the frontal and limbic regions for severe depression cases in contrast to milder ones. Such attention distribution patterns resonate with an established understanding of functional shifts associated with depression, as referenced in (Pizzagalli & Roberts, 2022; Johnston-Wilson et al., 2000; Lai et al., 2000). In sum, the attention maps acquired underscore BrainLM's adeptness at detecting clinically pertinent variations within functional networks. By illuminating the regions prioritized during the encoding process, the attention analysis enriches our interpretative capabilities, offering valuable neuroscientific insights. For a more comprehensive view of other clinical variables, refer to the supplementary Figure 8.

## 4.5 FUNCTIONAL NETWORK PREDICTION

We evaluated BrainLM's ability to segment parcels into intrinsic functional brain networks directly from fMRI activity patterns, without any network-based supervision. We organized parcels into 7 functional categories as delineated in previous cortical parcellation research. These groups

Table 3: Comparing methods for functional region identification. Parcels from 1000 UKB recordings were categorized into 7 regions without supervision. A kNN classifier on BrainLM's self-attention maps achieved the highest accuracy, outperforming alternatives using raw data and other representation learning techniques.

|                                            | Accuracy (%) |
| ------------------------------------------ | ------------ |
| BrainLM (attention weights)                | **58.8**     |
| Raw Data                                   | 39.2         |
| Variational Autoencoder (Kingma & Welling, 2013) | 49.4   |
| GCN (Kipf & Welling, 2016)                 | 25.9         |

encompassed visual, somatomotor, dorsal attention, ventral attention, limbic, frontoparietal, and default mode networks. For a held-out collection of 1,000 UKB recordings, we benchmarked various methodologies for classifying parcels into these 7 networks:

1) k-NN classifier using raw parcel time series data. 2) k-NN classifier leveraging parcel embeddings extracted from a variational autoencoder (VAE). This VAE, equipped with 3 encoding and 3 decoding layers, was trained to replicate time series while maintaining the same masking ratio and loss metrics as BrainLM. 3) k-NN classifier built on parcel embeddings from a 4-layer Graph Convolutional Network (GCN). This GCN was fine-tuned using a self-supervised masking-centric goal, aiming to master parcel representations with a 75% masking ratio. 4) k-NN classifier grounded on BrainLM's self-attention weights between each parcel token and its counterparts. For the classification process, we trained classifiers on 80% of the parcels for each recording, setting aside the rest 20% for evaluation. Notably, BrainLM's attention-driven approach distinctly surpassed the other methods, obtaining a parcel classification accuracy of 58.8% (refer to Table 3). In contrast, the k-NN classifier based on the GCN lagged, achieving only 25.9%.

These findings underscore BrainLM's strength in discerning functional brain topography purely from its pretraining phase. The self-attention maps provide meaningful insights about the network's identity, despite never encountering explicit labels during the training process.

## 5 DISCUSSION

This work presents BrainLM, the first foundation model for functional MRI analysis. By leveraging self-supervised pretraining on 6,700 hours of brain activity recordings, BrainLM demonstrates versatile capabilities for modeling, predicting, and interpreting human brain dynamics.

A key innovation lies in BrainLM's generalizable representations of fMRI recordings. The model achieves high accuracy in reconstructing masked brain activity sequences, even generalizing to held-out distributions. Furthermore, the model improves with an increased number of parameters showing that our approach scales with data and parameter size. This highlights the benefits of large-scale pretraining for learning robust encodings of spatiotemporal neural patterns. BrainLM also provides a powerful framework for biomarker discovery. By fine-tuning, brain dynamics can be decoded to predict clinical variables and psychiatric disorders better than baseline models. This could enable non-invasive assessment of cognitive health using resting-state fMRI alone. Finally, without any network-based supervision, BrainLM identifies intrinsic functional connectivity maps directly from pretraining, clustering parcels into known systems. This demonstrates how self-supervised objectives can extract organizational principles fundamental to the brain.

There remain exciting areas for future work. Multi-modal training could integrate fMRI with additional recording modalities, such as EEG and MEG, or different brain-wise information such as structural, functional, and genomic data. Probing a wider range of cognitive states and combined regularization may yield more generalizable representations. In future work, we could assess zero-shot classification on expanded functional atlases beyond the 7 networks used here. Overall, BrainLM provides a springboard for accelerated research at the intersection of neuroscience and AI. We hope that this work spurs further development of foundation models that can help elucidate the intricate workings of the human brain.

## 6 ACKNOWLEDGMENT

We thank the Yale Wu Tsai Institute for supporting JOC with a Postdoctoral Fellowship, the CAPES-Yale Graduate Scholars Program for support AHOF and are grateful to Insu Han and Amin Karbasi for their crucial insights and coding assistance in model training. Their contributions have significantly enhanced our research.

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
