# A APPENDIX

## A.1 SUPPLEMENTARY FIGURES

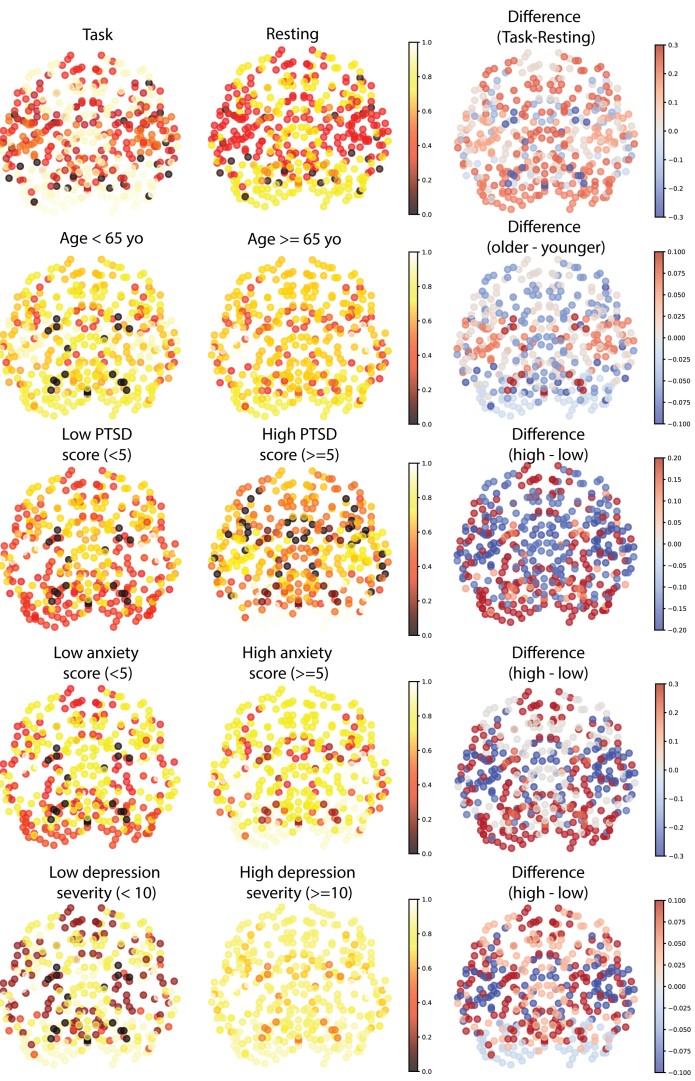

Figure 8: Attention visualization across 424 parcels. Each parcel is localized by its X and Y position and it is colored by the attention intensity with respect to the CLS token.

## A.2 Architecture Details

### A.2.1 Description of the model architectures used for clinical variable prediction

The architectural details of the models used for clinical variable prediction is given in Supplementary Table 4

Table 4: Architectural details for models used for regressing clinical variables

|  | Architecture | Parameters | Input Type |
|---|---|---|---|
| MLP | Enc: 3 FC layers, ReLU activation, Hidden layer dim = 512 | 46.2 M | Connectivity Matrix |
| GCN | Enc: (2 FC layer dim=256), | 2.3M | Parcels |
|  | Dec: (2 FC layer, dim=256) |  |  |
| LSTM | Enc: (1 LSTMcell, hid=400, 10 layers), | 14.6M | Parcels |
|  | Dec: (1 FC layer,dim=4000) |  |  |
| BrainLM (13M) | Enc: Transformer (layers=4, heads=4, dim=512, ff=1024), | 13M | Parcels |
|  | Dec: Transformer(layers=2, heads=4, dim=512, ff=1024) |  |  |
| BrainLM (111M) | Enc: Transformer (layers=12, heads=12, dim=768, ff=3072), | 111M | Parcels |
|  | Dec: Transformer(layers=8, heads=16, dim=512, ff=2048) |  |  |
| BrainLM (650M) | Enc: Transformer (layers=32, heads=16, dim=1280, ff=5120), | 650M | Parcels |
|  | Dec: Transformer(layers=8, heads=16, dim=512, ff=2048) |  |  |

### A.2.2 Description of the model architectures used for future state prediction

The architectural details of the models used for future brain state prediction is given in Supplementary Table 5

Table 5: Architectural details for models used for future brain state prediction

|  | Architecture | N of parameters |
|---|---|---|
| Neural ODE | Enc: (1 FC layer dim=450), | 1M |
|  | Hidden = (3 FC layers, dim=450) |  |
|  | Dec: (1 FC layer, dim=424) |  |
| LatentODE | ODE func: (3 layers,dim=1696), | 16.2M |
|  | Rec. RNN: (2 layers,hid=848), |  |
|  | Dec: (2 layers,dim=1696) |  |
| LSTM | Enc: (1 LSTMcell, hid=400, 10 layers), | 14.6M |
|  | Dec: (1 FC layer,dim=4000) |  |
| BrainLM (13M) | Enc: Transformer (layers=4, heads=4, dim=512, ff=1024), | 13M |
|  | Dec: Transformer(layers=2, heads=4, dim=512, ff=1024) |  |
| BrainLM (111M) | Enc: Transformer (layers=12, heads=12, dim=768, ff=3072), | 111M |
|  | Dec: Transformer(layers=8, heads=16, dim=512, ff=2048) |  |
| BrainLM (650M) | Enc: Transformer (layers=32, heads=16, dim=1280, ff=5120), | 650M |
|  | Dec: Transformer(layers=8, heads=16, dim=512, ff=2048) |  |

### A.2.3 Description of the model architectures used for functional area prediction

The architectural details of the models used for functional area prediction is given in Supplementary Table 6

Table 6: Architectural details for models used for functional area prediction

|  | Architecture | N of parameters |
|---|---|---|
| GCN | Enc: (2 FC layer dim=256), | 2.3M |
|  | Dec: (2 FC layer, dim=256) |  |
| VAE | Enc: (6 FC layer, hid=[512,256]), | 3.3M |
|  | Dec: (6 FC layer,dim=[256,512]) |  |
| BrainLM (13M) | Enc: Transformer (layers=4, heads=4, dim=512, ff=1024), | 13M |
|  | Dec: Transformer(layers=2, heads=4, dim=512, ff=1024) |  |

### A.2.4 TRAINING HYPERPARAMETERS

The training hyperparameters for different BrainLM models are reported in Supplementary Tables 7, 8, and 9.

Table 7: BrainLM (13M) Pretraining Hyperparameters

| PARAMETER | VALUE | PARAMETER | VALUE |
|---|---|---|---|
| Patch Size | 20 | Number of Timepoints per Voxel | 200 |
| Maximum Learning Rate | 1e-4 | Weight Decay | 1e-5 |
| Attention Dropout | 0.1 | Warmup Ratio | 0.05 |
| Total Batch Size | 256 | Gradient Clipping Max Norm | 1 |
| Optimizer | AdamW | Learning Rate Schedule | cosine annealing |

Table 8: BrainLM (111M) Pretraining Hyperparameters

| PARAMETER | VALUE | PARAMETER | VALUE |
|---|---|---|---|
| Patch Size | 20 | Number of Timepoints per Voxel | 200 |
| Maximum Learning Rate | 1e-4 | Weight Decay | 1e-5 |
| Attention Dropout | 0.0 | Warmup Ratio | 0.05 |
| Total Batch Size | 256 | Gradient Clipping Max Norm | 1 |
| Optimizer | AdamW | Learning Rate Schedule | cosine annealing |

Table 9: BrainLM (650M) Pretraining Hyperparameters

| PARAMETER | VALUE | PARAMETER | VALUE |
|---|---|---|---|
| Patch Size | 14x14 | Number of Timepoints per Voxel | 200 |
| Maximum Learning Rate | 1e-4 | Weight Decay | 1e-5 |
| Attention Dropout | 0.0 | Warmup Ratio | 0.05 |
| Total Batch Size | 256 | Gradient Clipping Max Norm | 1 |
| Optimizer | AdamW | Learning Rate Schedule | cosine annealing |

The training hyperparameters for the LatentODE models are reported in Supplementary Table 10.

Table 10: LatentODE Hyperparameters

| PARAMETER | VALUE | PARAMETER | VALUE |
|---|---|---|---|
| Optimizer | Adam | Batch Size | 32 |
| Learning rate | 1e-4 | | |

The training hyperparameters for the NODE models are reported in Supplementary Table 11.

Table 11: NODE Hyperparameters

| PARAMETER | VALUE | PARAMETER | VALUE |
|---|---|---|---|
| Optimizer | RMSprop | Learning rate | 1e-4 |
| Batch Size | 16 | | |

The training hyperparameters for the LSTM models are reported in Supplementary Table 12.

Table 12: LSTM Hyperparameters

| PARAMETER | VALUE | PARAMETER | VALUE |
|---|---|---|---|
| Optimizer | Adam | Learning Rate Schedule | ReduceLROnPlateau |
| Learning rate | 1e-3 | Factor | 0.5 |
| Batch Size | 16 | Minimum Learning Rate | 1e-6 |

Table 13: Performance comparison on the prediction of masked fMRI patches for BrainLM 13M. Shown is the coefficient of determination ($R^2$) between predicted and ground truth data for masked patches across various configurations of masking ratio (MR) and training data size. Rows correspond to models trained on either 1% or 100% of the UK Biobank dataset, while columns specify the masking ratios used during training and inference (0.2, 0.75, or 0.9). Overall, the models trained on the full dataset yielded better performance in the prediction of the masked patches, regardless of the masking ratio.

| Data size | MR=0.2 | MR=0.75 | MR=0.9 |
|---|---|---|---|
| 100 % | **0.464** | **0.352** | **0.241** |
| 1 % | 0.428 | 0.326 | 0.239 |

## A.3 SCALING LAW

### A.3.1 PREDICTION OF FUTURE BRAIN STATES

BrainLM also shows significant performance on future brain state prediction when finetuned on the whole original training set. Supplementary Table 16 shows the result of two large models when predicting future brain states. The increase in performance in the 650M parameter model vs. the 111M model further validates that performance benefits from scale.

Two different methods of future timepoint masking were used to accommodate the pretrained ViTMAE models' different patching scheme. The first, referred to as "overshoot", was done by masking all the patches that contained at least the last 20 timepoints of the recording. Since 20 timepoints were not exactly covered by whole multiples of patches, this meant that the fine-tuning task would entail predicting more than 20 timepoints. For the 111M model, which has a patch size of 16, that meant predicting 28 timepoints. For the 650M model, which has a patch size of 14, that meant predicting 23 timepoints. The second method, named "mixed", instead entailed zeroing timepoints to predict that were not wholly covered by the patches. Both of these methods were then evaluating on predicting 20 timepoints. Note how in both of these cases the task is more challenging than that of the original tokenization approach.

### A.3.2 ZEROSHOT METADATA REGRESSION

Zeroshot performance of BrainLM models on metadata variables regression is reported in Supplementary Table 17.

Table 14: Results for the regression of the clinical variables from learned latent space. Shown is the mean square error (MSE) between predicted and ground truth data for various model configurations. The columns indicate the regressed clinical variable. Rows indicate models trained on 1% or 100% of the UKB Biobank samples with different masking ratios (0.2, 0.75 or 0.9). The results show that the models trained on the full dataset yielded a more informative latent space for the regression of clinical variables, as evidenced by lower MSE values.

| | Data size | Age | PTSD (PCL-5) | Anxiety (GAD-7) | Neuroticism |
|---|---|---|---|---|---|
| MR=0.2 | 1 % | $0.836 \pm 0.015$ | $0.019 \pm 0.001$ | $0.093 \pm 0.006$ | $0.085 \pm 0.018$ |
| | 100 % | $\mathbf{0.826 \pm 0.047}$ | $\mathbf{0.018 \pm 0.001}$ | $\mathbf{0.090 \pm 0.005}$ | $\mathbf{0.081 \pm 0.004}$ |
| MR=0.75 | 1 % | $0.893 \pm 0.056$ | $0.026 \pm 0.004$ | $0.110 \pm 0.003$ | $0.110 \pm 0.006$ |
| | 100 % | $\mathbf{0.810 \pm 0.041}$ | $\mathbf{0.023 \pm 0.003}$ | $\mathbf{0.094 \pm 0.005}$ | $\mathbf{0.085 \pm 0.006}$ |
| MR=0.9 | 1 % | $0.919 \pm 0.057$ | $0.024 \pm 0.005$ | $\mathbf{0.094 \pm 0.003}$ | $0.086 \pm 0.004$ |
| | 100 % | $\mathbf{0.900 \pm 0.072}$ | $\mathbf{0.020 \pm 0.001}$ | $0.096 \pm 0.004$ | $\mathbf{0.084 \pm 0.006}$ |

Table 15: Performance comparison of model predictions on masked fMRI patches. Coefficient of determination ($R^2$) between predicted and ground truth data is reported across various model sizes and varying amounts of training data. Overall, reconstruction performance improves with both increased model size and data size, with a 3.76% improvement when scaling to the full amount of training data available in the UK Biobank.

| Data size | 111M parameters | 650M parameters |
|---|---|---|
| 0.1% | 0.0601 | 0.0673 |
| 1% | 0.1849 | 0.2219 |
| 10% | 0.1950 | 0.2218 |
| 50% | 0.2009 | 0.2232 |
| 100% | 0.2411 | **0.2567** |

Table 16: Future brain states prediction performance for the models fine-tuned on whole training set. Models labeled with "overshoot" were trained on predicting the last tokens corresponding to patches covering at least 20 timepoints. Models labeled "mixed" were trained on predicting the last tokens that corresponded to patches covering only masked timepoints, and timepoints that were not in these patches were zeroed prior to input in the model. All were evaluated on prediction of the last 20 timepoints. The 13M parameter model is reported as a reference and was trained according to the scheme described in the main text.

| | UKB | | | HCP | | |
|---|---|---|---|---|---|---|
| | $R^2$ | $R$ | $MSE$ | $R^2$ | $R$ | $MSE$ |
| BrainLM (13M, from Table 2) | 0.086 | 0.280 | 0.648 | 0.028 | 0.185 | 0.568 |
| BrainLM (111M) – overshoot | 0.094 | 0.307 | 0.020 | 0.057 | 0.241 | 0.018 |
| BrainLM (111M) – mixed | 0.093 | 0.305 | 0.020 | 0.057 | 0.246 | 0.018 |
| BrainLM (650M) – overshoot | **0.103** | **0.321** | **0.019** | 0.061 | 0.252 | 0.018 |
| BrainLM (650M) – mixed | 0.102 | 0.320 | 0.019 | **0.067** | **0.263** | **0.018** |

Table 17: MSE results for zeroshot metadata SVM regression. In the section titled "Mean Pool", recording embeddings were extracted by mean pooling of the encoder's token embeddings. In the 111M and 650M models marked with "no pad", the token embeddings corresponding only to padding values were removed prior to pooling. In the section titled "CLS", the CLS token of each recording was used in the regression.

| | Model | Age | PTSD (PCL-5) | Anxiety (GAD-7) | Neuroticism |
|---|---|---|---|---|---|
| **Mean Pool** | BrainLM (13M) | **0.697 ± 0.069** | 0.026 ± 0.008 | 0.122 ± 0.017 | 0.098 ± 0.017 |
| | BrainLM (111M) | 0.863 ± 0.049 | 0.025 ± 0.003 | 0.097 ± 0.015 | 0.077 ± 0.008 |
| | BrainLM (111M) – no pad | 0.875 ± 0.056 | 0.023 ± 0.006 | 0.096 ± 0.007 | 0.080 ± 0.014 |
| | BrainLM (650M) | 0.869 ± 0.049 | **0.022 ± 0.005** | 0.101 ± 0.009 | 0.079 ± 0.011 |
| | BrainLM (650M) – no pad | 0.859 ± 0.059 | 0.024 ± 0.005 | **0.093 ± 0.008** | **0.073 ± 0.006** |
| **CLS** | BrainLM (13M) | 0.922 ± 0.190 | **0.021 ± 0.003** | 0.156 ± 0.097 | 0.103 ± 0.019 |
| | BrainLM (111M) | **0.858 ± 0.064** | **0.021 ± 0.005** | 0.106 ± 0.015 | **0.080 ± 0.016** |
| | BrainLM (650M) | 0.876 ± 0.043 | 0.026 ± 0.010 | **0.095 ± 0.011** | 0.086 ± 0.019 |