# OpenReview forum: "BrainLM: A foundation model for brain activity recordings"
_ICLR.cc/2024/Conference — ICLR 2024 poster_

### Official Review · Reviewer_xyph · 2023-10-19

**Soundness:** 3 good
**Presentation:** 3 good
**Contribution:** 3 good
**Rating:** 6
**Confidence:** 3

**Summary:**

The authors train a neural network on fMRI recordings of brain activity: similarly to Large Language Models, the network is trained to predict missing values (random or future) with a self-supervised loss. The paper empirically demonstrated that the learned representations can be used in a variety of downstream tasks, even on a different dataset of fMRI recordings (HCP).

**Strengths:**

The performance metrics (R and R squared) reported by the authors are impressive given the variability of brain responses across subjects. This ability of the learned representations to generalize beyond the training set (UKB) to a new dataset with new subjects (HCP) is impressive.

**Weaknesses:**

A standard protocol for evaluating representations "downstream" is to fine-tune a linear predictor or "probe" on top of the representations [1, 2]. The fact that the fine-tuned predictor is linear is important: it can measure how well the representations "cluster" [3], or linearly correlate, with downstream labels. Yet in this paper, the fine-tuned predictor is a nonlinear neural network with three layers. This raises doubts on whether the representations actually correlate with downstream labels (e.g. age), if they need to undergo many nonlinear transformations before the linear prediction layer. Could the authors comment on this?

The authors use the terminology of "foundation model" and "language model" which suggest a novel approach to analyzing fMRI recordings; yet, at its core, their model is a neural network trained with a self-supervised loss (e.g. predict random or future missing values) on fMRI recordings. And there is a rich and existing literature of relevant works that is worth acknowledging. Consider for example [4, 5] as well as their own reference sections. With this context, could the authors explain what contributions they make that are novel with respect to the current literature?

[1] Alain and Bengio, Understanding intermediate layers using linear classifier probes, ICLR workshop 2017.

[2] van den Oord et al. Representation Learning with Contrastive Predictive Coding. Preprint 2019.

[3] Wang et al. Understanding Contrastive Representation Learning through Alignment and Uniformity on the Hypersphere. ICML, 2020.

[4] Caucheteux et al. Deep language algorithms predict semantic comprehension from brain activity. Nature scientific reports, 2022.

[5] Pasquiou et al. Neural Language Models are not Born Equal to Fit Brain Data, but Training Helps. ICML, 2022.

**Questions:**

I believe the authors make interesting contributions and would be willing to raise my score if my concerns and questions (in the weaknesses section) are addressed.

---

> ### Author Response · Authors · 2023-11-23
> **Response to Reviewer xyph**
>
> We thank the reviewer for their positive comments and constructive questions. We provide detailed responses below.
>
> Q: “A standard protocol for evaluating representations "downstream" is to fine-tune a linear predictor or "probe" on top of the representations [1, 2]. The fact that the fine-tuned predictor is linear is important: it can measure how well the representations "cluster" [3], or linearly correlate, with downstream labels. Yet in this paper, the fine-tuned predictor is a nonlinear neural network with three layers. This raises doubts on whether the representations actually correlate with downstream labels (e.g. age), if they need to undergo many nonlinear transformations before the linear prediction layer. Could the authors comment on this?”
>
> R: We agree with the reviewer that a more complex decoder can facilitate the task. However, we note that non-pretrained models are also included that serve as a baseline for predicting the clinical variables. BrainLM outperforms this baseline, showing that the unsupervised pretraining helps learn a good embedding. If the embedding learned by the model during self-supervised pretraining was suboptimal, then the non-pretrained baseline would be equally good as BrainLM, but this is not the case. As such, we can conclude that the non-linear predictor is not the primary factor driving the improved performance. We are open to incorporating results using a linear predictor if the reviewer deems it necessary, however we have no reason to believe that this will change the results qualitatively.
>
> Q:”The authors use the terminology of "foundation model" and "language model" which suggest a novel approach to analyzing fMRI recordings; yet, at its core, their model is a neural network trained with a self-supervised loss (e.g. predict random or future missing values) on fMRI recordings. And there is a rich and existing literature of relevant works that is worth acknowledging. Consider for example [4, 5] as well as their own reference sections. With this context, could the authors explain what contributions they make that are novel with respect to the current literature?”
>
> R: We appreciate the reviewer for highlighting two relevant pieces of literature. There is an important distinction between our work and the studies presented in [4,5]. The primary focus of these papers was on employing self-supervised learning for a specific task: identifying areas in the cortex that have high correlation to GPT-2 or other large language models. Specifically, these studies observed that language processing areas were better explained by these models. This brain-scoring approach has proven successful in understanding the information used by these language models for their tasks and how they relate to different brain areas. It has also been applied to non-language areas like V1 and IT.
>
> However, it's important to note that none of these models were explicitly trained to model the spatiotemporal dynamics of brain recordings. Furthermore, they did not evaluate the ability of these models to generalize to the prediction of clinical variables, a distinctive feature of our work. We will incorporate these studies into the related works section and provide a stronger comparison against their approach.

---

### Official Review · Reviewer_mZsL · 2023-10-26

**Soundness:** 3 good
**Presentation:** 4 excellent
**Contribution:** 3 good
**Rating:** 6
**Confidence:** 5

**Summary:**

The paper presents an transformer architecture based model to (a) reconstruct fMRI sequences and (b) predict clinical variables. As fMRI recordings can be seen as multivariable sequences with rich temporal and cross-variable dependencies, such application has a lot of merit and might give rise to a foundational model for fMRI data analysis.

The architecture of the model follows other built for similar purposes, and is trained to predict masked-out patches of fMRI BOLD sequences. The model is trained on a large corpus of data from UK BioBank, the largest fMRI data repository currently available.

BrainLM shows performance superior to several benchmarks (but: see a question about those below).

**Strengths:**

The paper is written very clearly, one can follow it easily, the material is presented logically, clearly, and at the same time with sufficient amount of details.

I wholeheartedly support the motivation behind his work. Indeed a LLM-based approach to fMRI sequence analysis makes perfect sense, this methodology should yield great results and help fMRI community to overcome multiple issues with heterogeneity of the data, allowing to pull together large dataset enabling us to train powerful models for neuroscience research and clinical application.

The way the authors approach the problem is in my opinion technically sound and their solution is exactly what one might expect to see in this context. This gives me a lot of confidence in the soundness of their results and future applicability of the BrainLM model.

**Weaknesses:**

Table 3: In the section on related work you mention some models that were trained on fMRI to narrowly predict just one clinical variable, and, presumably, those models are state of the art in their respective predictions of clinical variable. However, in Table 3 you seem to compare BrainLM against some other results. My question here I guess whether the comparison presented in the table reflects the state of the art achieved by other models in predicting certain clinical variables, and not just state of the art using a certain method. The fact that an LSTM or an SVM performs poorly is not that interesting to see, what would be interesting to see is how much better or worse performance we get when compared to the state of the art method specifically trained to predict GAD-7 for example. Same for other clinical scores, a methods does not have to predict all of them at once. (If, in fact, the methods you use in this comparison are the current state of the art, please let me know that this is the case and I will withdraw this criticism)

Table 3: Are the "competitor" results obtained by you training these methods on the same data, or these are the numbers reported in the respective studies by the authors of those methods?

The worry I have in those questions above is whether the benchmarks presented are the "strong" ones.

**Questions:**

Table 1: It appears that allowing the model to see more context (less masking) yields significantly better better performance. (a) Have you tried to train the model with even less masking (MR=0.1) and what were the results? (b) If the trend of "less masking = better performance" continues, then what would be the limit of this: at what point using less masking will start to hurt generalizability?

Table 2: It would help the reader a lot, if, in addition to MSE and R^2 values, you would also report the absolute error in the original units of measurement. For general clinical variables such as age it would help the general public to better understand what kind of error the model makes as measured in years (which is immediately understandable to anyone). For clinicians working on GAD, PHQ, etc scales it would also be immediately impressive to see the performance estimate directly in those units.

Page 5 last paragraph: When you were fine-tuning BrainLM to regress metadata variable, did you predict all of the listed variables (age, neuroticism, PTSD, anxiety) at once as 4-head output, or it was a separate fine-tuning process per variable?

Table 3: What does the "Raw data" row indicate? For example for "Age" we have "2.0 +/- 0.22"? Does this somehow show the variability of age in the dataset (unlikely as 2.0 is too small number for age variability)? Or is this the error that a naive linear model makes? Unclear.

Figure 4: "CLS" abbreviation is not explained anywhere in the text.

---

> ### Author Response · Authors · 2023-11-23
> **Response to Reviewer mZsL**
>
> We thank the reviewer for their positive and constructive comments. We provide detailed responses below.
>
> Q: “Table 3: In the section on related work you mention some models that were trained on fMRI to narrowly predict just one clinical variable, and, presumably, those models are state of the art in their respective predictions of clinical variable. However, in Table 3 you seem to compare BrainLM against some other results. My question here I guess whether the comparison presented in the table reflects the state of the art achieved by other models in predicting certain clinical variables, and not just state of the art using a certain method.”
>
> R: To clarify, the referenced MAE models trained on fMRI are only focused on stimuli-decoding based on visual areas. None of the models have been used for clinical variable prediction. In the literature, common approaches for clinical variable prediction involve the use of SVM and MLPs to correlation matrix input [1]. These are the models we have included in Table 2 (previous Table 3).
>
> [1] Vieira, Sandra, Walter HL Pinaya, and Andrea Mechelli. “Using deep learning to investigate the neuroimaging correlates of psychiatric and neurological disorders: Methods and applications.” Neuroscience & Biobehavioral Reviews 74 (2017): 58-75.
>
> Q: “Table 3: Are the "competitor" results obtained by you training these methods on the same data, or these are the numbers reported in the respective studies by the authors of those methods?”
>
> R: We obtained the performance of competing methods ourselves. The methods designed for clinical variable prediction were trained on other datasets, preventing us from incorporating their performance directly. Furthermore, we requested access to their code for comparison but received no response. As a result, we constructed our own baselines based on the information available in their manuscript. We did extensive hyperparameter search for each method for fairness and presented the best performance of each method in our analysis.
>
> Q: “Table 1: It appears that allowing the model to see more context (less masking) yields significantly better better performance. (a) Have you tried to train the model with even less masking (MR=0.1) and what were the results? (b) If the trend of "less masking = better performance" continues, then what would be the limit of this: at what point using less masking will start to hurt generalizability?”
>
> R: Regarding generalization, the models with lower masking ratio (and thus more information) have an easier prediction task. As such, their in-distribution performance is higher. However, when tested with other masking ratios, their performance decreases significantly. As a result, for all subsequent comparisons, we opted to use the 75% masking model, which exhibits the best generalization among all models.
> Q: “Page 5 last paragraph: When you were fine-tuning BrainLM to regress metadata variable, did you predict all of the listed variables (age, neuroticism, PTSD, anxiety) at once as 4-head output, or it was a separate fine-tuning process per variable?”
>
> R: All models were fine-tuned separately. We will clarify this in the manuscript.
>
> Q: “Table 3: What does the "Raw data" row indicate? For example for "Age" we have "2.0 +/- 0.22"? Does this somehow show the variability of age in the dataset (unlikely as 2.0 is too small number for age variability)? Or is this the error that a naive linear model makes? Unclear.”
>
> R: This is the error of a linear model. We will clarify this in the manuscript.
>
> Q: Figure 4: "CLS" abbreviation is not explained anywhere in the text.
>
> R: Thank you for pointing this out. The CLS token is a reference to a “CLASS” token, which is used for classification/regression after pretraining. This token effectively acts as an embedding which captures the whole dynamics. We have clarified this in the manuscript.

---

### Official Review · Reviewer_NWMt · 2023-11-01

**Soundness:** 2 fair
**Presentation:** 3 good
**Contribution:** 3 good
**Rating:** 6
**Confidence:** 4

**Summary:**

The paper describes a new model for fMRI data trained in a self-supervised manner and then fine-tuned to predict a variety of downstream tasks of interest in neuroscience including brain activity forecasting, clinical outcome prediction and brain network identification. The model is shown to enable better performance than a variety of reported baselines on the above tasks.

**Strengths:**

The paper operates at the intersection of two areas of intense interest (large-scaled foundation models and neuroscience), and it situates itself well in both domains with a clear setup and motivation, and clear illustration of the modeling paradigm chosen. The results look impressive relative to the baselines chosen.

**Weaknesses:**

My main concerns lie with the fact that the network is very modestly sized for a foundation model, and that the baselines aren't a priori motivated in a way that would support the contributed model as truly SOTA.

Regarding the former, 13M parameters is smaller than the MLP, LatentODE, and LSTM baselines, and much smaller than what I would consider to be even a small foundation model. For example, HuBERT-Base is a 90M parameter model trained on ~1K hours of speech data (12 layers and 8 heads with larger dimensions than BrainLM). So while BrainLM is a model trained on a nontrivial amount of fMRI data using SSL, it is remarkably small for a foundation model. That would be fine if performance were good relative to a set of baselines chosen based on prior results on fMRI decoding, but I'm not sure if this were the case. At least, the paper does not make it obvious (for example, the NODE, LatentODE, and GCN citations point back to the original papers, none of which have neuroscience applications). Thus, the baselines are also novel, so it's unclear how impactful beating them should be. Considering that the UKB data was released in 2016 and the HCP data published in 2021, I would be surprised if there are no baselines in the literature against which the present contribution can be evaluated. Without this contextualization against past work, neither the performance numbers nor having trained a small SSL model on thousands of hours of fMRI data is sufficiently impactful for ICLR in my judgement.

Some additional comments:
* In discussion of large-scale SSL for neuroscience, the paper would benefit from also considering Kostas et al. 2021 10.3389/fnhum.2021.653659.
* More sophisticated methods could be used for interpretability than attention maps (e.g. Chefer et al. CVPR 2021).
* The legend for Fig. 5 has LatentODE, LSTM, and NODE whereas the text talks about ODENets and non-pretrained BrainLM. Please add the non-pretrained BrainLM results or correct the text. In addition, the text could make a clearer mapping from legend to citations (e.g. "NODE (Chen et al.), LatentODE (Rubanova rt al.)"). Finally, I am surprised that there is an unpretrained BrainLM baseline in table 4 but not table 3.
* Something strange seems to be going on with NODE training -- it performs much worse than other methods and gives a large *negative* $R^2$ value in table 4 (likely indicating severe model mismatch, and giving me concern about whether training failed entirely for this model).
* Considering the claims regarding "springboard for accelerated research" via a foundation model, the paper should make explicit that weights and code will be released.

**Questions:**

* Was figure 3 randomly sampled or cherry-picked?
* The generalization numbers ($R^2=0.464$ within dataset or $.278$ across) need to be contextualized -- how do we know they are good?
* Why are the 1% and 100% training results of interest? Should we not be surprised that more data improves performance, and if it is surprising why not show a trend over more than just two data sizes?
* 40% dropout seems quite high -- is this correct, or a typo?
* Stats on fig 5: how were they done? Hard to imagine BrainLM beats LSTM and LatentODE at later timepoints given the visualized error bars, was it something like BrainLM vs the average of the rest such that the effect of NODE dominated?
* What motivated the specific network architectures chosen? They vary substantially in number of parameters.
* What were the other training details (e.g. learning rates and their schedules, dropout and other regularization for the SSL training, gradient clipping if any, etc)? These would be fine in an appendix but should be included for reproducibility.

---

> ### Author Response · Authors · 2023-11-23
> **Response to Reviewer NWMt**
>
> We thank the reviewer for their constructive comments and questions. We provide detailed responses below.
>
> Q: “BrainLM is a model trained on a nontrivial amount of fMRI data using SSL, it is remarkably small for a foundation model.”
>
> R: We appreciate the reviewer's perspective on the comparative scale of our model, especially when juxtaposed with the larger foundation models prevalent in other domains. Taking this constructive feedback into account, we have expanded our research scope and developed enhanced versions of our model, now featuring 111 million and 650 million parameters, respectively. This represents a substantial increase, up to 50 times the parameter count of our initial model.
>
> The effectiveness of these larger models is evident in the revised Tables 1, 2 and Supp. Table 17, as well as in Figure 4 of our manuscript, which collectively demonstrate a significant boost in performance attributable to the increased parameter count. However, we were not able to finish training the 650M parameter model in Table 1 due to time constraints. In a future version of the manuscript, this model will be included.
>
> In line with our commitment to collaborative research and open science, we plan to make all variants of these models accessible to the broader research community. By sharing these models, we aim to foster further exploration and innovation in the field, inviting other researchers to build upon, refine, or even challenge our findings.
>
> Q: Comparison to baselines.
>
> R: We acknowledge the reviewer's concern regarding the challenge of evaluating results in clinical variable prediction. We conducted a thorough literature review and found that the use of Support Vector Machines (SVM) or Multi-Layer Perceptrons (MLPs) trained on correlation matrices remains the predominant approach in the field (see references [1-3]). This prevalence influenced our decision to include these methods in our study.
>
> Moreover, the absence of more recent methodologies in existing literature led us to develop our own baseline models for comparison. We ensured fairness in evaluation by allocating an equal amount of hyperparameter optimization effort to all models presented in our study.
>
> We are open to suggestions from the reviewer regarding the inclusion of additional methods. If there are specific, newer approaches that the reviewer believes would enrich our comparative analysis, we are more than willing to incorporate these into our study. Our commitment is to provide a comprehensive and robust benchmarking that reflects the current state of the art in the field.
>
> [1] Woo, Choong-Wan, Luke J. Chang, Martin A. Lindquist, and Tor D. Wager. “Building better biomarkers: brain models in translational neuroimaging.” Nature Neuroscience 20, no. 3 (2017): 365-377.
> [2] Vieira, Sandra, Walter HL Pinaya, and Andrea Mechelli. “Using deep learning to investigate the neuroimaging correlates of psychiatric and neurological disorders: Methods and applications.” Neuroscience & Biobehavioral Reviews 74 (2017): 58-75.
> [3] Orru, Graziella, William Pettersson-Yeo, Andre F. Marquand, Giuseppe Sartori, and Andrea Mechelli. “Using support vector machine to identify imaging biomarkers of neurological and psychiatric disease: a critical review.” Neuroscience & Biobehavioral Reviews 36, no. 4 (2012): 1140-1152.
>
> Q: “In discussion of large-scale SSL for neuroscience, the paper would benefit from also considering Kostas et al. 2021 10.3389/fnhum.2021.653659.”
>
> R: We thank the reviewer for the reference. It has now been included in the related work section.
>
> Q: “The legend for Fig. 5 has LatentODE, LSTM, and NODE whereas the text talks about ODENets and non-pretrained BrainLM. Please add the non-pretrained BrainLM results or correct the text. In addition, the text could make a clearer mapping from legend to citations (e.g. "NODE (Chen et al.), LatentODE (Rubanova rt al.)"). Finally, I am surprised that there is an unpretrained BrainLM baseline in table 4 but not table 3.”
>
> R: We thank the reviewer for pointing out the inconsistency between the legend and the text. We have addressed the issue. We have also added the citations to the methods used in the table. Regarding the absence of a non-pretrained BrainLM model, due to the time constraints associated with this deadline, our efforts were primarily concentrated on experiments pertaining to model scaling. Consequently, we were unable to train the BrainLM 13M model from scratch for clinical variable regression within the stipulated time frame. However, we recognize the importance of this aspect and plan to include it in a future iteration of the manuscript. This addition will undoubtedly provide a more comprehensive understanding of the model's capabilities and limitations.

---

> > ### Author Response · Authors · 2023-11-23
> > **Part 2**
> >
> > Q:”Something strange seems to be going on with NODE training -- it performs much worse than other methods and gives a large negative R2 value in table 4 (likely indicating severe model mismatch, and giving me concern about whether training failed entirely for this model).”
> >
> > R: We agree with the reviewer's observation regarding the problematic nature of the results when employing the Neural ODE (NODE) model on the fMRI dataset. During the training phase, we shared similar concerns about NODE's performance with this specific type of data. In response, we undertook extensive experimentation to enhance its effectiveness.
> >
> > Despite our rigorous attempts, we concluded that NODE's inherent structural design might not be optimally configured for the unique properties of our fMRI data. We speculate that the substantial dimensionality coupled with the intrinsic noise prevalent in fMRI datasets poses formidable challenges for NODE, limiting its capability to accurately represent and analyze such intricate data.
> > We remain open to suggestions and would eagerly incorporate any modifications or alternative approaches recommended by the reviewer to enhance our methodological framework.
> >
> >
> > Q: “Considering the claims regarding "springboard for accelerated research" via a foundation model, the paper should make explicit that weights and code will be released.”
> >
> > R: We agree with the reviewer and we are committed to ensuring the accessibility of our work. In line with this, we plan to release all models and accompanying code concurrent with the publication of our manuscript. To facilitate ease of access and use within the research community, we will host the models on Hugging Face. This decision aligns with our goal of promoting transparency and fostering collaborative research efforts. We have added a statement about this to the manuscript.
> >
> >
> > Q: “Was figure 3 randomly sampled or cherry-picked?”
> >
> > R: The example showcased in Figure 3 was selected through a random sampling process. This approach was adopted to ensure the representativeness of the data presented. The robustness of our model is reflected in the high R2 values obtained in the prediction tasks for both UK Biobank (UKB) and Human Connectome Project (HCP) datasets. These values are indicative of a strong fit and suggest that a similarly high level of accuracy would be observed across other randomly sampled recordings.
> >
> >
> > Q:: Why are the 1% and 100% training results of interest? Should we not be surprised that more data improves performance, and if it is surprising why not show a trend over more than just two data sizes?
> >
> > R: The new Figure 4 and tables in your manuscript effectively demonstrate the scalability of the BrainLM models. By including models trained on varying data sizes (0.1%, 1%, 10%, 50%, and 100%) and showcasing models with different parameter sizes, we provide a comprehensive view of how model performance scales with data size and complexity. This approach not only reinforces the notion that larger models yield better performance but also offers valuable insights into the efficiency and effectiveness of our BrainLM models across different training scenarios.
> >
> > Q: 40% dropout seems quite high -- is this correct, or a typo?
> > R: We thank the reviewer for spotting the typo. We used 10% attention dropout. This has been corrected in the manuscript.
> >
> > Q: Stats on fig 5: how were they done? Hard to imagine BrainLM beats LSTM and LatentODE at later timepoints given the visualized error bars, was it something like BrainLM vs the average of the rest such that the effect of NODE dominated?
> > R: For each comparison between BrainLM and the models LSTM, LatentODE, and NODE, statistical significance was calculated. We've adopted the convention of marking an asterisk in our figures whenever BrainLM's performance significantly surpasses all three models in these pairwise comparisons. This methodology, along with a detailed explanation, will be included in the figure's description to ensure a comprehensive understanding of the comparative performance metrics.
> >
> > Q: What motivated the specific network architectures chosen? They vary substantially in number of parameters.
> >
> > R: Our objective was to encompass a broad spectrum of potential models, with a specific focus on those that have demonstrated efficacy in modeling spatiotemporal dynamics, including LatentODE, GCN, NODE, and LSTM. Given the novelty of applying these methods to fMRI data, as we previously noted, it was imperative for us to establish our own benchmarks for baseline comparison. This approach not only facilitates a comprehensive understanding of each model's capabilities in this new context but also lays the groundwork for future explorations in the field.

---

> > > ### Author Response · Authors · 2023-11-23
> > > **Part 3**
> > >
> > > Q: What were the other training details (e.g. learning rates and their schedules, dropout and other regularization for the SSL training, gradient clipping if any, etc)? These would be fine in an appendix but should be included for reproducibility.
> > >
> > > R: We agree with the reviewer, this should have been included. We have now included it in the supplementary material (section A.2).

---

### Official Review · Reviewer_4kZC · 2023-11-01

**Soundness:** 2 fair
**Presentation:** 3 good
**Contribution:** 3 good
**Rating:** 6
**Confidence:** 4

**Summary:**

In this paper, the authors propose BrainLM, a foundation model designed to analyze fMRI brain activity recordings. BrainLM is trained on 6,700 hours of fMRI data using a Transformer-based architecture. Experimental results show that the proposed BrainLM generalizes well to diverse downstream tasks. Through fine-tuning, it can accurately predict clinical variables such as age and anxiety, while in zero-shot learning, it can identify intrinsic brain networks from fMRI data, offering an approach to understanding spatiotemporal brain dynamics.

**Strengths:**

The paper is written clearly and is well-structured; the motivation is intuitive and the method is easy to follow.
The author collected a large-scale fMRI dataset, which enhances the reliability and generalizability of the pre-trained BrainLM.
Extensive experiments on diverse downstream tasks like clinical variable prediction and brain state prediction are relatively comprehensive. The additional attention analysis also covers the spatial patterns in the brain.

**Weaknesses:**

Lack of novelty, the MAE-like pretrain model and method used in this paper are reasonable, but they are known in the field.

Lack of comparative baselines: The baselines used for both clinical variable prediction and brain state prediction are somewhat limited. Baselines such as SVM are outdated. The unsupervised MAE-like model mentioned in the related work should also be considered for comparison.

Besides random masking and future masking, more mask strategies should be investigated, like uniform masking and parcel-level (spatial) masking.

**Questions:**

What is the computational cost of the pretraining?

---

> ### Author Response · Authors · 2023-11-23
> **Response to Reviewer 4kZC**
>
> We thank the reviewer for their positive comments. We provide detailed responses below.
>
> Q: “Lack of novelty, the MAE-like pretrain model and method used in this paper are reasonable, but they are known in the field.”
>
> R: We acknowledge the reviewer's observation that MAE-like pretraining is recognized in the field and well-established in the domains of computer vision and natural language processing. However, the use of such architecture for modeling neural dynamics in whole-brain fMRI data is an area yet to be fully explored. Previous studies predominantly focused on voxel-based recordings and specific brain areas. The voxel-based approach has inherent limitations, such as restricted spatial coverage due to the large number of voxels per area. Our study diverges from this tradition by applying the MAE framework to parcelated fMRI data, an approach not previously undertaken. Our work marks the first attempt to leverage parcel-based dynamics to model the whole brain.
>
> Moreover, our novelty extends to the comprehension of the model. By evaluating its proficiency in predicting functional regions and demonstrating a correlation between brain-wide attention and relevant clinical variables, we underscore the practical utility of these foundational models in the context of fMRI recordings.
>
> In summary, while MAEs have been explored before in other contexts, their application to extensive fMRI data at this extreme scale, coupled with the sophisticated data engineering involved and the myriad of fine-tuning applications demonstrated, marks a significant departure from existing methodologies. We believe this work firmly establishes the potential of foundation models as a paradigm shift for brain imaging analysis.
>
> Q: “Lack of comparative baselines: The baselines used for both clinical variable prediction and brain state prediction are somewhat limited. Baselines such as SVM are outdated.The unsupervised MAE-like model mentioned in the related work should also be considered for comparison.”
>
> R: Regarding the comparison to previous work, the reviewer rightfully points out that SVM and MLP may be considered somewhat outdated. However, such approaches remain widely employed within the neuroscience community for assessing brain function and characterizing disorders [1-3]. Given the limited exploration of machine learning models for predicting clinical variables, we intentionally included these methods in our study. Despite conducting an extensive literature review, no additional relevant approaches were identified. We are receptive to incorporating any additional methods suggested by the reviewer.
>
> Furthermore, the reviewer references other fMRI MAE models mentioned in the related work section, which specifically focus on the visual cortex for decoding stimuli presented to subjects. We believe these models are too domain-specific for our current setting. Nevertheless, we are open to the idea of training our own versions of these models using visual-only information to provide a comparative analysis, if the reviewer deems this necessary.
>
> [1] Woo, Choong-Wan, Luke J. Chang, Martin A. Lindquist, and Tor D. Wager. “Building better biomarkers: brain models in translational neuroimaging.” Nature Neuroscience 20, no. 3 (2017): 365-377.
>
> [2] Vieira, Sandra, Walter HL Pinaya, and Andrea Mechelli. “Using deep learning to investigate the neuroimaging correlates of psychiatric and neurological disorders: Methods and applications.” Neuroscience & Biobehavioral Reviews 74 (2017): 58-75.
>
> [3] Orru, Graziella, William Pettersson-Yeo, Andre F. Marquand, Giuseppe Sartori, and Andrea Mechelli. “Using support vector machine to identify imaging biomarkers of neurological and psychiatric disease: a critical review.” Neuroscience & Biobehavioral Reviews 36, no. 4 (2012): 1140-1152.
>
> Q: “Besides random masking and future masking, more mask strategies should be investigated, like uniform masking and parcel-level (spatial) masking.”
>
> R: We appreciate the reviewer's suggestion. Our choice of random masking is informed by prior research in the literature, where it has proven to be an effective method for pretraining MAE-like models [1]. However, we acknowledge that alternative masking techniques, such as parcel-level masking, could be a suitable alternative, potentially comparable to higher masking ratios like 80% or 90%, given the extensive number of brain parcels. In fact, we have previously experimented with parcel (spatial) masking but did not observe an improvement and as such we did not further explore this direction. We are interested in exploring this, and other, masking techniques in future work.
>
> [1] Feichtenhofer, Christoph, Yanghao Li, and Kaiming He. "Masked autoencoders as spatiotemporal learners." Advances in neural information processing systems 35 (2022): 35946-35958.
>
> Q: What is the computational cost of the pretraining?
>
> Currently, any given model can be pretrained on the UKBioBank data in around 6 days on an A100 GPU.

---

### Author Response · Authors · 2023-11-23
**Response to all reviewers**

We thank the reviewers for their insightful and constructive feedback. Based on the comments, we have made substantial improvements to both our manuscript and models. Alongside our individual responses to each reviewer, we wish to highlight the major changes and improvements made in our revised submission:

- Expansion of Model Size: We have significantly increased the capacity of our models from the original 13M parameters to 111M and 650M parameters.
- Demonstration of Scaling Behavior: With these models we show strong scaling behavior of performance with respect to model size and data size, as shown in Figure 4.
- Superior Performance Metrics: The larger models have significantly improved performance in self-supervised and fine-tuning tasks, including future state prediction and both zero-shot and fine-tuned clinical variable prediction (see Tables 1 and 2, and Supp. Table 17).

We believe that these models, which boast 50 times more parameters than our original model and are trained on the most extensive fMRI dataset available to date, represent a significant stride forward in modeling brain activity recordings. We are excited to share these foundational models with the neuroscience and machine learning communities via GitHub and HuggingFace, and hope that they will serve as foundational tools for future research.

---

> ### Comment · Reviewer_NWMt · 2023-11-23
>
> Dear authors, I am unable to see your individual responses to me (or any other reviewers). Please double check the visibility of the postings. AC, I wonder if you can see the responses or change their visibility? Thank you.

---

> > ### Author Response · Authors · 2023-11-23
> >
> > Dear reviewer, we have added our responses to each of the reviewers. Perhaps there was a delay in uploading. We believe our comments should be visible now. Please let us know if they are not.

---

### Meta-Review · Area_Chair_eQ6u · 2023-12-09

**Metareview:**

This paper proposes BrainLM, a novel Transformer-based foundation model for analyzing fMRI brain activity recordings. Trained on a massive 6,700-hour dataset, BrainLM demonstrates impressive generalizability across various downstream tasks. Fine-tuning enables accurate prediction of clinical variables like age and anxiety, while zero-shot learning facilitates identification of intrinsic brain networks, paving the way for deeper understanding of spatiotemporal brain dynamics. The proposed model outperforms existing baselines, establishing itself as a promising tool for fMRI data analysis.

The paper excels in its clear research motivation, accessible methodology, and use of a large-scale dataset to train the BrainLM model. Extensive experimentation across diverse tasks and impressive results compared to existing baselines solidify the effectiveness of BrainLM. The attention analysis further adds value by revealing insights into brain activity patterns. This work bridges the gap between large-scale foundation models and neuroscience, paving the way for exciting advancements in the field.

The reviewers have raised concerns regarding its small size, unclear comparisons with existing benchmarks, and lack of novelty. Additionally, the evaluation methods for clinical variable prediction and representation quality require further justification. Addressing these issues is crucial for establishing BrainLM's significance and contribution to fMRI analysis.

**Justification For Why Not Higher Score:**

The paper excels in its clear research motivation, accessible methodology, and use of a large-scale dataset to train the BrainLM model. Extensive experimentation across diverse tasks and impressive results compared to existing baselines solidify the effectiveness of BrainLM. The attention analysis further adds value by revealing insights into brain activity patterns. This work bridges the gap between large-scale foundation models and neuroscience, paving the way for exciting advancements in the field.

**Justification For Why Not Lower Score:**

The reviewers have raised concerns regarding its small size, unclear comparisons with existing benchmarks, and lack of novelty. Additionally, the evaluation methods for clinical variable prediction and representation quality require further justification. Addressing these issues is crucial for establishing BrainLM's significance and contribution to fMRI analysis.

---

### Decision · Program_Chairs · 2024-01-16

Accept (poster)